# Health Coaching-Based Interventions for Oral Health Promotion: A Scoping Review

**DOI:** 10.3390/dj11030073

**Published:** 2023-03-06

**Authors:** Remus Chunda, Peter Mossey, Ruth Freeman, Siyang Yuan

**Affiliations:** School of Dentistry, University of Dundee, Dundee DD1 4HR, UK

**Keywords:** health coaching, motivational interviewing, oral health promotion, intervention, behaviour change, health education, person-centred care, oral health, communication

## Abstract

Background: Health coaching-based interventions can support behaviour change to improve oral health. This scoping review aims to identify key characteristics of health coaching-based interventions for oral health promotion. Methods: The Preferred Reporting Items for Systematic reviews and Meta-Analyses extension for Scoping Reviews checklist and the Joanna Briggs Institute manual for evidence synthesis were used in this review. A search strategy using medical subject heading terms and keywords was developed and applied to search the following databases: CINAHL, Ovid, PubMed, Cochrane Library and Scopus. Thematic analysis was used to synthesise the data. Results: Twenty-three studies met the inclusion criteria and were included in this review. These studies were predominantly based on health coaching and motivational interviewing interventions applied to oral health promotion. The following are the characteristics of health coaching-based interventions extracted from themes of the included studies: (a) Health professionals should be trained on the usage of motivational interviewing/health coaching interventions; (b) oral health professionals should acquire motivational techniques in their practice to engage patients and avoid criticisms during the behaviour change process; (c) routine brief motivational interviewing/health coaching intervention sessions should be introduced in dental clinics; (d) traditional oral health education methods should be supplemented with individually tailored communication; and (e) for cost-effectiveness purposes, motivational interviewing/health coaching strategies should be considered. Conclusions: This scoping review reveals that health coaching-based techniques of health coaching and motivational interviewing can significantly impact oral health outcomes and behaviour change and can improve oral health professional–patient communication. This calls for the use of health coaching-based techniques by dental teams in community and clinical settings. This review highlights gaps in the literature, suggesting the need for more research on health coaching-based intervention strategies for oral health promotion.

## 1. Introduction

Behaviour change interventions can improve oral health and lead to success in the treatment of oral diseases [1,2]. Evidence suggests that sustainable behaviour change will not be achievable by just providing oral health information to the patient [3]. Patients are the key decision-makers concerning their oral health behaviour change. Change must be generated proactively by individual patients themselves if it is to be achievable. When a clinician provides oral health instruction to the patient, the patient is assumed to act on the information given and expected to change their behaviours [4]. However, this approach to patient education and communication has proved not to be effective [5]. For this reason, alternative psychological interventions for evoking behaviour change have been proposed, explored and developed in the improvement of oral health. One way of assisting patients in changing and maintaining their oral health behaviours is for the dental health professionals to adopt a person-centred approach such as health coaching-based interventions (HCBIs) which utilise a motivational interviewing (MI) technique. The National Health Service (NHS) in England has defined health coaching (HC) as a person-centred behaviour change communication skill [6]. The person-centred approach to HC facilitates mutual interaction between the health professional and the subject empowering the latter to pro-actively change behaviours that impact on their own individual health and wellbeing [7]. Croffoot et al. [8] defined motivational interviewing (MI) as a person-centred approach for increasing internal motivation, while Curtin et al. [9] described MI as a therapeutic skill for supporting patients’ readiness to change and assisting them to commit to the change process. Both MI and HC underpin Carl Rogers’s person-centred care (PCC) philosophy [10] which is vital for behaviour change.

Evidence suggests that clinicians who utilise HC-based intervention strategies (HCBIs), such as motivational interviewing and health coaching approaches, often yield positive health outcomes in their patients [11]. MI and HC skills offer clear information and communication for self-care and self-management. These skills serve as platforms for continuity of care, smooth transition and fast access to reliable health advice. In addition, MI and HC skills emphasise empathy, respect and support for patients and their families or caregivers [12], focus on holistic care and the integration of oral health with general health and wellbeing, while achieving sustainable health improvement [13].

The significance of HCBIs is that these interventions have supported patients in building their own capability to change behaviours and maintain good health [14]. Proponents of these strategies point out that HCBIs help in the improvement of health outcomes through decreasing of health care costs, improvement of listening abilities, the building of dedicated care teams and development of goals which are specific, measurable, achievable, realistic and timely for both patients as well as clinicians [15,16,17].

Despite the fact that health coaching uses motivational interviewing skills to encourage patients for behaviour change, to our knowledge, it is not clear in terms of the scope of evidence for interventions using health coaching strategies in the dental literature. Although a few review studies [18,19,20] have explored the role of motivational interviewing technique in promoting oral health, the characteristics of health coaching-based interventions have never been systematically examined in the dental literature. While systematic reviews are used for identifying and retrieving “particular questions and to appraise and synthesise the results of this search to inform practice, policy and future research”, a scoping review has been chosen to be suitable given its focus on the “scope” of the available literature to have an overview of the research evidence pertinent to a specific topic [21].

This scoping review aimed to determine the key characteristics of HCBIs for oral health promotion among adult patients in the community and clinical settings. In this review HCBIs refers to MI or HC strategies.

## 2. Materials and Methods

The Preferred Reporting Items for Systematic reviews and Meta-Analyses extension for Scoping Reviews checklist (PRISMA-ScR) [22] and the Joanna Briggs Institute evidence synthesis manual [23] was used in this review, where population, concept and context were used to identify the research question and eligibility criteria.

Population: Adult patients aged 18 years or over.Concept: Health coaching-based interventions for oral health promotionContext: Community and clinical settings

### 2.1. Eligibility Criteria

The JBI framework was used to develop the inclusion and exclusion criteria for this review (Table 1) to aid in the selection of appropriate papers. The studies included in this review were published articles, qualitative, quantitative, or mixed methods and peer-reviewed papers of an oral health intervention study that involved HC or MI to promote oral health.

### 2.2. Search Strategy

A search strategy based on medical subject heading (MeSH) terms was formulated in collaboration with the librarian (SM) and was used to search the following electronic databases: Cochrane Library, Ovid, Scopus, CINAHL and Medline via PubMed. The MeSH terms strategies represented five broad themes: communication, health coaching and motivational interviewing, oral health, intervention and behaviour change. The MeSH term-based search strategy was as follows: {[(Coach*)]} OR {[“Motivational Interviewing”(Mesh)] AND [“Oral Health”(Mesh)]} OR {[“Health Education, Dental”(Mesh)] OR [“Oral health”] OR [“Oral Hygiene”(Mesh)]} AND {[Intervention OR Improv* OR Program* OR Promotion*] OR [“Health Promotion”(Mesh)] OR [“Pilot Projects” (Mesh)]}. MeSH terms generated studies from 1997 to 2020.

### 2.3. Data Selection

Following initial electronic database search, data (articles) which were found to have met the eligibility criteria were exported in groups by database, cross-referenced by EndNote X9 (Clarivate, London, UK), where duplicates were removed. Additional articles were also searched manually in appropriate journals, reference lists and Google Scholar, which were also added to EndNote X9. All the combined studies were exported from EndNote X9 to RAYYAN (Qatar Computing Research Institute (QCRI, Doha, Qatar), where blinding was applied for the first and second reviewer (RC and SY) to make their selection decisions. Upon discussion, SY and RC agreed to include studies that were blindly selected through RAYYAN. Then, the PRISMA-ScR (Figure 1) was used for data selection.

### 2.4. Quality Assessment, Data Charting and Synthesis

Some authors note that one of the differences between a scoping review and a systematic review is the lack of an appraisal of the scoping review to assess the quality of the included studies [24]. However, others have pointed out that while it is not a requirement for a scoping review to methodologically assess the quality of included studies and determine the extent to which the study has addressed a possibility of bias in its design [21,25], a critical appraisal of the included studies is necessary. Additionally, Pham et al. [24] observe that it is often the lack of quality appraisal in scoping reviews which has been frequently cited as a limitation when comparing them to systematic reviews, and this has resulted in them being interpreted as less rigorous. Some authors have further argued that it is the lack of quality assessment in scoping reviews that makes interpretation of results difficult [26]. As a result, some authors have argued that that scoping reviews can assess study quality even though it is not always a requirement [24]. These authors observe however that quality assessment should not be used as key for inclusion criteria when selecting articles for scoping reviews. It is also imperative for researchers to have clear objectives before conducting a scoping review study as the main purpose of a scoping review is to identify and map available evidence in the literature [27,28]. In this scoping review quality assessment of the included studies was carried out using the JBI’s critical appraisal checklist tools (Table 2) which assisted in assessing the relevance, trustworthiness and checking results of published papers [29]. Included studies were thus grouped based on their design and the checklist questions from the JBI appraisal tools were used to assess the quality of the studies see Table 3 below. After screening the included studies for quality, they were rearranged into new groups to show the effects as “high”, “medium” and “low” quality (Table 3). Eighteen studies were of high quality, three of medium quality and two of low quality. Please note that these results were not used to determine study inclusion.

On data charting, a data extraction form adapted from the JBI synthesis manual [23] was used to draw out basis information about each study and the main variations considered appropriate to provide key information on the analysis of characteristics of HCBIs for oral health promotion (Table 4). The charted information included author, year of publication, country of origin, aim of the study, type of sample, methodology used, intervention and summary of key findings (Table 3). The extracted recommendations and identified themes/characteristics of each of the included studies were compiled in groups (Table 5) below created with the JBI evidence synthesis manual [23].
dentistry-11-00073-t002_Table 2Table 2Quality assessment checklist of the included articles adapted from JBI [29].Type of StudyJBI Critical Appraisal Checklist Questions UsedSystematic reviews and research synthesesIs the review question clearly and explicitly stated?Were the inclusion criteria appropriate for the review question?Was the search strategy appropriate?Were the sources and resources used to search for studies adequate?Were the criteria for appraising studies appropriate?Was critical appraisal conducted by two or more reviewers independently?Were there methods to minimise errors in data extraction?Were the methods used to combine studies appropriate?Was the likelihood of publication bias assessed?Were recommendations for policy and/or practice supported by the reported data?Were the specific directives for new research appropriate?Randomised Control TrialsWas true randomisation used for assignment of participants to treatment groups?Was allocation to treatment groups concealed? Were treatment groups similar at the baseline?Were participants blind to treatment assignment? Were those delivering treatment blind to treatment assignment?Were outcomes assessors blind to treatment assignment?Were treatment groups treated identically other than the intervention of interest?Was follow-up complete and if not, were differences between groups in terms of their follow up adequately described and analysed?Were participants analysed in the groups to which they were randomised?Were outcomes measured in the same way for treatment groups? Were outcomes measured reliably?Was an appropriate statistical analysis used?Was the trial design appropriate, and any deviations from the standard RCT design (individual randomisation, parallel groups) accounted for in the conduct and analysis of the trial?Quasi-experimental/Interventional StudiesIs it clear in the study what is the cause’ and what is the ‘effect’ (i.e., there is no confusion about which variable comes first)?Were the participants included in any comparisons similar?Were the participants included in any comparisons receiving similar treatment/care, other than the exposure or intervention of interest?Was there a control group?Were there multiple measurements of the outcome, both pre and post the intervention/exposure?Was follow-up complete and if not, were differences between groups in terms of their follow up adequately described and analysed?Were the outcomes of participants included in any comparisons measured in the same way?Were outcomes measured reliably?Was an appropriate statistical analysis used?
dentistry-11-00073-t003_Table 3Table 3Quality assessment results of the included studies.Type of StudyStudyQualityRCTGodard et al. [30]MediumRCTStenman et al. [31]HighRCTLópez-Jornet et al. [32]HighSystematic Review and Research Syntheses Cascaes et al. [33]MediumRCTBrand et al. [34]HighQuasi-experimental/Interventional studyJohansson et al. [35]HighQuasi-experimental/Interventional studySaengtipbovorn and Taneepanichskul [36]HighSystematic Review and Research SynthesesYevlahova and Satur [37]HighQuasi-experimental/Interventional studyTellez et al. [38]MediumSystematic Review and Research SynthesesKay et al. [1]HighSystematic Review and Research SynthesesGao et al. [39]HighRCTNaidu et al. [40]HighQuasi-experimental/Interventional studyCinar et al. [13]HighQuasi-experimental/Interventional studyDermen et al. [41]HighSystematic Review and Research SynthesesWerner et al. [2]HighRCTTellez et al. [42]HighRCTStenman et al. [43]HighRCTJönsson et al. [44]HighRCTAlrashdi et al. [45]HighSystematic Review and Research SynthesesWatt [46]LowQuasi-experimental/Interventional study Cinar et al. [47]HighRCTRigau-Gay et al. [48]HighSystematic Review and Research SynthesesAntoniadou and Varzakas [49]Low
dentistry-11-00073-t004_Table 4Table 4Data charting.Author(s)YearCountry of OriginAimStudy SampleMethodInterventionKey FindingGodard, Dufour, Jeanne [30]2011FranceTo assess the effectiveness of MI vs. Conventional basic instruction on compliance with plaque control and periodontitisN = 51 patientsQuantitativeMI on OH improvementMI can be used to target and modify inappropriate behaviour and can be implemented into a periodontal treatment plan Stenman, Wennström, Abrahamsson [31]2018SwedenTo evaluate whether a single MI session can influence periodontal therapy. Evaluate retention of OH behavioursN = 26 patients QuantitativeSingle session MI on periodontal healthA single MI session could not be proven to be of the long-term beneficial additive effect regarding prevention of relapse in oral hygiene behaviourNaidu, Nunn, Irwin, [40]2015TrinidadTo compare MI to oral health education on oral health knowledge, attitudes, beliefs and behavioursN = 79 parents and caregiversQuantitativeCompare MI to traditional HE, behavioursUse of MI during oral health information giving session had positive effect on parent/caregiver oral health knowledge, attitudes and behaviours compared to traditional oral health education.Alrashdi, Hameed, Mendez, Farokhi [45]2020USATo assess the effectiveness of oral health education and behavioural intervention on improving knowledge, attitudes and behaviours of refugee families using MI. N = 66 parentsQuantitativeBehaviour change, MI oral health improvementBehavioural intervention with oral health education did not improve oral health-related knowledge, attitudes, or behaviours in a diverse group of recent refugee families.López-Jornet, Fabio, Consuelo, Paz [32]2014SpainTo assess the effectiveness of MI-behavioral skills for plaque control in patients with hyposalivationN = 60 patientsQuantitativeOral hygiene practices, MI, OH improvementIn patients with hyposalivation, application of OHI based on cognitive principles and MI offers benefits for periodontal healthCascaes, Bielemann, Clark, Barros [33]2014BrazilTo analyse the effectiveness of MI at improving oral hygiene habitsN = 78 articlesSystematic ReviewMI, OH improvementMore research needed to fully understand the impact of MI on oral health and understand counselling interventions.Cinar, Oktay, Schou [47]2014TurkeyTo evaluate the impact of HC compared to health education on oral health and diabetes management among patients with type 2 Diabetes N = 186 patientsQuantitativeCompare health education to health coaching on oral health improvement and behaviour changeHC has a significantly higher impact on better management of diabetes and oral health when compared to formal health education.Brand, Bray, MacNeill, Catley, Williams [34]2013USATo evaluate whether brief MI is effective in improving internal motivation for oral hygiene behaviour.N = 56 patientsQuantitativeSingle session MI on oral hygiene behaviour change and internal motivationOne-time MI session is insufficient for improving oral hygiene in long-standing maintenance patients.Johansson, Torge, Lindmark [35]2020SwedenTo examine the feasibility of HC in a nursing home to improve oral health careN = 106 nursing home staff and residentsQuantitativeHC on oral health improvementHC can support nursing staff in maintaining a high level of oral health among residents.Saengtipbovorn, Taneepanichskul [36]2015ThailandTo assess if Lifestyle Change plus Dental Care can improve oral health knowledge, attitude and practice in diabetic patientsN = 132 patientsQuantitativeOral hygiene practices, MI, behaviour change and oral health improvementA combination of lifestyle change with dental care improves knowledge, attitude and practice toward oral health and diabetes mellitus in the elderly with Type 2 DiabetesYevlahova, Satur [37]2009AustraliaTo identify and assess the effectiveness of behaviour models as a basis for individual oral health promotionN = 32 studiesSystematic ReviewMI, behaviour change and oral health improvementThere is a need to develop an effective model for chairside oral health promotion. MI has the potential to be developed within the oral health fieldTellez, Virtue, Neckritz, Bhoopathi, Hernández, Shearer [38]2019USATo examine fidelity of individual MI intervention and assess baseline characteristics related to older adults’ self-efficacy oral health-related quality of lifeN = 60 patientsQuantitativeTailored MI and OHWsFindings from the study support the reliability of one-on-one MI intervention for conducting oral health education Kay, Vascott, Hocking, Nield [1]2016UKTo review evidence regarding the use of MI in general dental practice so practitioners can decide whether it might be an important skill to develop within their practices.N = 20 studiesSystematic Revieworal health workers, MI as a skill for professionalsMI based on the concept of autonomy support has potential for helping patients with poor oral health. MI training for dental personnel could be a beneficial skill.Gao, Lo, Kot, Chan [39]2014ChinaTo synthesise the evidence on the effectiveness of MI compared with conventional (health) education (CE) in improving oral healthN = 221 recordsSystematic Review MI vs. standard HE in oral health improvementMI shows varied success in improving oral health. Potential of MI in dental health care, like improving periodontal health, remains controversial. Additional studies are needed to understand the role of dental practice on a bigger scale.Watt [46]2010UKTo identify models for health behaviour, change and to evaluate evidence for their effectivenessN = 32 studiesNarrative ReviewMI and Dental SettingMI interventions were found to be the most effective method for altering health behaviours in a clinical setting.Cinar, Freeman and Schou [13]2018Turkey & DenmarkTo assess the effectiveness of HC vs. health education using clinical and type 2 diabetes patients in Turkey and Denmark.302 patientsN = 186 participants TurkeyN = 116 participants DenmarkQuantitativeHC on Behaviour changeHC has a greater impact on type 2 diabetes managementDermen, Ciancio, Fabiano [41]2014USATo provide initial evidence that, compared with a didactic control intervention, a brief MI-based intervention (BMI) delivered by dental practitioners can yield greater oral health improvementsN = 60 in-patientsQuantitativeBrief MI on OHA brief intervention using MI methods can be delivered by dental professionals and has potential utility for promoting improved oral hygiene. Werner, Hakeberg, Dahlström, Eriksson, Sjögren, Strandell, Svanberg, Svensson, Boman [2]2016SwedenTo study the effectiveness of psychological interventions in adults and adolescents with poor oral healthN = 843 articles in 2013N = 378 articles in 2015Systematic review MI on oral hygiene practicesStatistically significant differences reported in favour of psychological interventions in oral health behaviour and self-efficacy in toothbrushing. Tellez, Myers Virtue, Neckritz, Lim, Bhoopathi, Hernandez, Ismail [42]2020USATo assess the treatment fidelity of an individual-based oral health education intervention using MI compared to group-based oral health educationN = 180 patientsQuantitativeTailored MI compared to HEThis study supports the fidelity of this intervention and the improvement of all non-clinical outcomes after 12 months amongst the MI groupStenman, Lundgren, Wennström, Ericsson, Abrahamsson, [43]2012SwedenTo evaluate the effect of a single session of MI on self-performed periodontal infection control.N = 44 patientsQuantitativeSingle session MI on improving periodontal infection controlA single freestanding MI session had no significant effect on the individuals’ standard of self-performed periodontal infection control in a short-term perspective.Jönsson, Öhrn, Lindberg, Oscarson [44]2010SwedenTo evaluate the effectiveness of individually tailored oral health education compared with standard oral health education on periodontal health. N = 113 patientsQuantitativeIndividually tailored oral health education using MI on periodontal health improvementAn individually tailored oral health educational program intervention, in combination with scaling is preferable to the standard oral hygiene education in non-surgical periodontal treatment.Rigau-Gay, Claver-Garrido, Benet, Lusilla-Palacios, Ustrell-Torrent [48]2020SpainTo evaluate the effectiveness of a single session of MI on enhancing oral hygiene among orthodontic patients compared with conventional education aloneN = 130 patientsQuantitativeSingle session MI with orthodontic patientsShort time-MI combined with conventional education is useful to improve oral hygiene, since it decreases plaque and gingival indexes, in adolescents and young adults wearing fixed appliances. Antoniadou, Varzakas [49]2020GreeceTo identify diet and oral health coaching methods and models for the independent elderlyN/ANarrative ReviewHC on diet and oral health/dietary habitsDental and other medical professionals should re-evaluate their roles as health coaches to improve dietary habits and nutritional intake of their patients
dentistry-11-00073-t005_Table 5Table 5Extracted recommendations and identified themes/characteristics. StudyRecommendationsThemes/CharacteristicsWerner et al. [2]Interventions may be used if benefits and risks, cost-effectiveness and ethical aspects are consideredMI/HC strategies are cost-effectiveTellez et al. [42]Training, dental personnel on MI approach would make it a powerful alternative particularly in settings where there are no dentists.Train health professionals to provide MI/HC in clinical settingsProvide individually tailored communicationStenman et al. [43]Dental professionals should learn how to motivate the patients. MI should be added to traditional health education methods.Train health professionals to provide MI/HC in clinical settingsSupplement traditional health education methods with tailored communicationIntroduce brief routine MI/HC sessionsJönsson et al. [44]Non-surgical periodontal treatment is achieved through individually tailored oral health communication. The treatment programme is possible for special trained dental hygienists to perform in their treatment of patients with chronic periodontitis.Provide individually tailored communication.Train health professionals to provide MI/HC in clinical settingsAlrashdi et al. [45]Health education should be supplemented with other interventions to achieve positive oral health outcomesSupplementing traditional health education methods with tailored communicationCinar et al. [47]Training of health professionals Engagement of health workers and their patients on the behaviour change journeyTrain health professionals to provide MI/HC in clinical settingsEngage clients or patients and avoid negative feedback and criticismProvide individually tailored communicationRigau-Gay et al. [48]Training dental hygienists as part of the working team to become experts in MI focused on dental settings could be useful, for self-empowerment and motivation.Train health professionals to provide MI/HC in clinical settingsProvide Brief routine MI/HC sessionsAntoniadou & Varzakas [49]Quick health education sessions followed by tailored feedback using empathy is important in terms of helping elderly adults in improving their oral health. Health professionals should re-evaluate their role as health coaches.Train health professionals to provide MI/HC in clinical settingsProvision of individually tailored communicationEngage clients or patients and avoid negative feedback and criticismBrief routine MI/HC sessionsMI/HC strategies are cost-effectiveDermen et al. [41]A brief MI intervention delivered by dental professionals is feasible and efficacious than didactic methods.Introduction of brief routine MI sessionsTrain health professionals to provide MI/HC in clinical settingsCinar et al. [13]There is a need for health promotion strategies with health coaching for the management of type 2 diabetes that focus on multidisciplinary approaches including oral health. The study highlights the importance of (i) training of health professionals on HC and (ii) patient-health professional collaboration.Train health professionals to provide MI/HC in clinical settingsEngage clients or patients and avoid negative feedback and criticismProvide individually tailored communicationMI/HC strategies are cost-effectiveWatt [46]There is potential to further develop the MI approach within the oral health field.Train health professionals to provide MI/HC in clinical settingsGao et al. [39]A growing interest of dental professionals in MI and suggests some the potential of applying MI in oral healthTrain health professionals to provide MI/HC in clinical settingsProvide individually tailored communicationKay et al. [1]Build relationships between OHPs and patients. Training of OHPs in MI techniques. Avoidance of negative judgements Train health professionals to provide MI/HC in clinical settingsEngage clients or patients and avoid negative feedback and criticismTellez et al. [38]MI is reliable for HE. Engaging patients with a menu of options rather than a prescribed checklist Provide individually tailored communicationEngage clients or patients and avoid negative feedback and criticismTrain health professionals to provide MI/HC in clinical settingsYevlahova & Satur [37]MI has the potential to be developed within the oral health field.Train health professionals to provide MI/HC in clinical settingsSaruta Saengtipbovorn & Taneepanichskul [36]Knowledge and education are essential for engaging clients in making behaviour or lifestyle change Engage clients or patients and avoid negative feedback and criticismJohansson et al. [35]Extended support from dental care, including practical training and feedback on an individual basis, could benefit nursing staff in providing oral health careTrain health professionals to provide MI/HC in clinical settingsBrand et al. [34]Adoption of MI in dental and dental hygiene education. Adopt evidence-based patient-engagement strategies.Train health professionals to provide MI/HC in clinical settingsEngage clients or patients and avoid negative feedback and criticismIntroduction of brief routine MI/HC sessionsCascaes et al. [33]Beneficial effects of MI conducted individually in clinical settings. MI appears to be a promising approach for changing individual behaviourTrain health professionals to provide MI/HC in clinical settingsBrief routine MI/HC sessionsLópez-Jornet et al. [32]Dental professionals must provide instructions on oral hygiene, education and information and identify obstacles to behavioural improvementProvide individually tailored communicationEngage clients or patients and avoid negative feedback and criticismNaidu et al. [40]MI approach had a positive effect compared to traditional health education. Development of the person-centred counselling approachProvide individually tailored communicationSupplementing traditional health education methodsStenman et al. [31]A single MI session does not add beneficial effects to standard periodontal therapy for efficient oral hygiene behaviourIntroduction of brief routine MI/HC sessionsGodard et al. [30]Dentists are recommended to start using MI. This procedure could be applied in similar clinical settings when the patient’s active role is crucial to treatment success.Train health professionals to provide MI/HC in clinical settingsProvide individually tailored communication


## 3. Results

A total of (*n* = 238) articles were retrieved from the five electronic databases namely Ovid (*n* = 45); PubMed (*n* = 85); Cochrane Library (*n* = 64); Scopus (*n* = 37); and CINAHL (*n* = 7). An extra group of (*n* = 9) articles was manually searched from key journals and reference lists and was added to (*n* = 238) articles making a total of (*n* = 247) articles. After exclusion, full access articles included in this scoping review were (*n* = 23) Figure 1.

### 3.1. Study Characteristics

All the included studies were from the past 10 years, except one that was from 1997. Most studies (*n* = 13) were from Europe while the rest were from the USA, China, Australia, Brazil, India and Trinidad. Majority of studies (*n* = 16) used quantitative designs, while (*n* = 7) were reviews.

There were (*n* = 6) studies that assessed the efficacy of HCBIs on periodontal health, while (*n* = 3) studies assessed the efficacy of HCBIs on OH management among diabetes melitus type 2 patients. Two (*n* = 2) studies assessed the effect of HBCIs on oral health knowledge of parents and care givers of children. Two studies (*n* = 2) assessed the effect of HCBIs on dietary habits. There were (*n* = 10) studies that assessed the effect of HCBIs on poor OH and OHI.

### 3.2. Key Findings in Relation to Characteristics/Themes

The results were based on themes/characteristics synthesised from the 23 articles that were the focus of this review to answer the research question.

#### 3.2.1. Engage Clients or Patients and Avoid Negative Feedback and Criticism

Evidence in this review shows that clinicians, including oral health professionals (OHPs), should always endeavour to engage their patients and strive to avoid criticising or giving negative feedback during behaviour change communication. Cinar et al. [13] and Cinar et al. [47] stressed the need for patient-health professional engagement in the behaviour change journey using the HC approach. Tellez et al. [38] noted that health professionals should engage patients by providing them with a menu of options rather than a prescribed checklist when using MI technique. In a similar analysis, Brand et al. [34] recommended adopting evidence-based patient engagement strategies through MI. Likewise, López-Jornet et al. [32] observed that OHPs should engage with their patients or clients by identifying obstacles to behavioural improvement. Similarly, Antoniadou and Varzakas [49] pointed out that oral healthcare provision in elderly patients is affected by many factors. Therefore, OHPs should re-evaluate their roles by adopting HC because it is a better communication option in terms of better engagement of patients in the healthcare delivery process. Kay et al. [1] reported that avoidance of negative judgements in patients with poor oral hygiene helps build a therapeutic relationship between clinicians and their patients, which is necessary for successful oral health improvement in dental surgery. Similarly, Saengtipbovorn and Taneepanichskul [36] expressed the need to engage patients with the knowledge and education required for behaviour or lifestyle changes.

#### 3.2.2. Supplement Traditional Health Education Methods with Tailored Communication

Most of the included studies stressed the need to use individually tailored health education messages instead of traditional HE. Jönsson et al. [44] observed that tailored HE adapted to individual goals and problems improved oral hygiene behaviour and periodontal health. Similarly, Antoniadou and Varzakas [49] pointed out that HC sessions should be performed, followed by tailored feedback. Likewise, Cinar et al. [47] and Cinar et al. [13] recommend using HC-tailored communication in the management of the oral health of patients with type 2 diabetes. This was reiterated by Tellez et al. [42] and Tellez et al. [38], who stressed on the need for a tailored individual-based HE intervention using the MI technique instead of conventional HE. Similarly, Gao et al. [39] recommend using tailored communication in the form of an MI counselling approach to change oral health behaviours. Naidu et al. [40] reported that tailored communication using the MI approach during oral health information delivery had a positive effect on toothbrushing behaviour, oral health knowledge and oral health fatalism. Similarly, Alrashdi et al. [45] observed that HE should be supplemented with other interventions to achieve positive oral health outcomes. Gordard et al. [30] also noted that a tailored MI technique compared to conventional basic instruction for improving compliance with plaque control among patients with periodontitis is effective. Similarly, López-Jornet et al. [32] observed that HE based on MI offers benefits to periodontal health in patients with hyposalivation; however, they pointed out that long-term studies are required.

#### 3.2.3. Introduce Brief Routine MI/HC Sessions

Dermen et al. [41] recommended introducing brief MI intervention sessions delivered by dental professionals as feasible and efficacious strategies compared to didactic methods. Similarly, Antoniadou and Varzakas [49] observed that brief HC sessions improved oral health. This has been also seen in Rigau-Gay et al.’s work [48] that a short brief MI session combined with traditional HE performed routinely effectively improved oral hygiene. However, another study [43] noted that single freestanding MI sessions had no significant effects on self-performed periodontal infection control measures in the short term. Similarly, Stenman et al. [31] also found a single session of MI was not effective in preventing periodontal disease relapse in the long term. This was reiterated by Brand et al. [34], who observed that a single MI session is insufficient for improving oral hygiene in longstanding maintenance patients. Nevertheless, Cascae et al. [33] pointed out that MI has beneficial clinical effects when applied individually. These differences in results could be attributed to the study settings, design, or methodology.

#### 3.2.4. Train Health Professionals to Provide MI/HC in Clinical Settings

There were seventeen of the included studies which reiterated on the need of training of health professionals in HC or MI skills to enhance their work. Rigau-Gay et al. [48] noted the need to train dental hygienists in the management of MI. Similarly, Kay et al. [1] observed that MI training could be an instrumental skill for dental professionals. This is reiterated by Antoniadou and Varzakas [49], who recommended that dental and other medical professionals be trained in HC. Similarly, Dermen et al. [41] observed that MI methods delivered by dental professionals have the potential to improve oral health, whereas Jönsson et al. [44] recommended training hygienists to tailor messages. Likewise, Brand et al. [34] explained the need for adopting MI strategies in dental hygiene education. Similarly, Watt [46] observed that MI interventions are the most effective method for altering health behaviours in clinical settings. Nevertheless, Gao et al. [39] noted that the potential for using MI in dental and oral healthcare remains controversial; they suggested conducting more studies to understand its roles. Tellez et al. [38] recommended teaching dental professionals to adopt MI strategies in one-on-one-patient communication, whereas Tellez et al. [42] recommended training para-dental professionals working in areas without dentists to adopt MI approaches. Stenman et al. [31] observed that dental professionals should be taught how to motivate patients, whereas Yevlahova and Satur [37] noted the need to develop an effective MI model for chairside oral health promotion. Similarly, Johansson et al. [35] advocated the use of HC to support and train nursing staff in maintaining a high level of oral health among residents. Cinar et al. [13] and Cinar et al. [47] also stressed the need for training health professionals on the HC strategy, while Godard et al. [30] observed that dentists should start using the MI strategy. Cascaes et al. [33] noted the beneficial effects of MI in a clinical setting.

#### 3.2.5. Cost-Effectiveness of MI/HC Strategies

Werner et al. [2] noted that behavioural change interventions may be useful in dental treatment if cost-effectiveness is a factor. These authors pointed out that this might benefit the staff and planners of dental care. Similarly, Cinar et al. [13] made the point that HC might well be found to be an extremely cost-effective model considering the potential for prevention as opposed to therapeutic health care interventions that deal with the consequences of disease occurrence. Antoniadou and Varzakas [49] also noted that HC could be one of the most cost-effective ways to achieve positive health outcomes before costly expenses such as financial costs or severe complications arise for both the healthcare system and patient.

## 4. Discussion

This scoping review aimed to map and synthesise evidence on health coaching-based interventions for oral health promotion with a focus on the key characteristics of the intervention programs. Our discussions are therefore organised based on the key themes/characteristics identified from the extracted findings.

### 4.1. Engage Clients or Patients and Avoid Negative Feedback and Criticism

The finding on engaging clients or patients and avoiding negative feedback and criticism [1,13,32,34,36,38,47,49] in this review mirrors the work of Wong-Rieger and Rieger [50], who observed that HC was essential for assisting and engaging patients in the behaviour change journey. These authors argue that health professionals should engage and support patients using HC to manage their health. However, Wong-Rieger and Rieger’s work lacked robustness in methodology, as it was a narrative review translated from French. Nevertheless, this study highlights the key role of patient centredness in HC strategies. An alternative view is mirrored by Freeman [51], who suggests that engaging and motivating patients to change oral health behaviour is a complex issue. She stressed the need to develop an understanding of the behaviour change journey among OHPs and patients and also to allow patients to explore their attitudes to both costs and benefits of changing behaviours. This will provide opportunities for OHPs to assess patients’ readiness to change behaviours. It is indeed true that usage of MI requires an understanding of the behaviour change model, so OHPs can engage with patients and assist them in achieving the long-term health goals of compliance and maintenance of newly acquired health behaviours. Notwithstanding every clinician should be aware that, the context to change individual behaviour is driven by intrinsic motivation which is a key element to MI; as such, all external factors become irrelevant if there is no engagement with the patient’s internal drive.

### 4.2. Supplement Traditional Health Education Methods with Tailored Communication

Supplementing traditional health education methods with tailored communication [13,30,32,38,39,40,42,44,45,49] was reflected in the work of Carter et al. [52], who suggested that HC is a useful approach for clinicians and their patients in many ways. These authors have found that since piloting the HC strategy in NHS England, it has produced positive health outcomes and behaviour changes given its function as a tailored communication platform for patients compared to traditional communication. However, it is worth noting that this study only involved NHS Trust in East England. Nonetheless, this study highlights important issues regarding the efficacy of the HC approach, which is consistent with the findings of this review. Furthermore, a study by Wanyonyi et al. [53] established the role played by tailored communication in oral health and patient behaviour change. These authors observe that the use of tailored communication is feasible and effective in creating behaviour change for patients in primary care settings. The findings from this review suggest that knowledge of patients’ personal characteristics, and the subsequent tailoring of health messages may result in positive health effects and the adoption of health behaviours by patients.

### 4.3. Introduce Brief Routine MI/HC Sessions

One of the findings of this review was the introduction of brief MI/HC sessions in dental practice settings [31,33,34,41,43,48,49]. This finding mirrors the work of Williams [54], who suggests that the use of brief MI/HC sessions in dental practice, whether 15–20 min or extended (multiple 45-min sessions), can be effective in oral health patient behaviour change practice. This finding might seem inconsistent and may take time for oral health professionals to assimilate it; however, it could be vital for oral health improvement. Additionally, the application of MI/HC techniques in the management of oral diseases is still in its infancy, indicating that more studies are required for definitive conclusions in this area. Nonetheless, this finding is critical as it has pointed out the potential of MI in oral disease management and in the subsequent patient’s oral health behaviour change journey.

### 4.4. Train Health Professionals to Provide MI/HC in Clinical Settings

HCBIs involving MI or HC have not been widely implemented in many clinical scenarios to assist patients in changing their oral health behaviours. In addition, it is also unclear which category of people is most beneficial. However, the findings in this review on training of health professionals to provide MI/HC [1,30,31,33,34,35,37,38,39,41,42,44,46,47,48,49] are vital and mirror the work of other scholars [55,56,57,58]. For instance, Kopp et al. [55] like all the other scholars particularly found that training dental professionals to provide MI in a clinical setting can be effective in the patient behaviour change process. It is indeed important that dental health professionals should be trained in MI and HC strategies up to a proficiency level to ensure the efficacy of the MI and HC approaches in oral health promotion sessions at a clinical setting.

### 4.5. Cost-Effectiveness for MI/HC Strategies

Another useful finding of this review was the cost-effectiveness of HCBIs using MI and HC strategies [2,13,49]. This finding reflects previous studies which suggested that indeed behaviour change interventions of HC and MI can be cost-effective in lifestyle and behavioural disease management [52,59,60]. To our knowledge, there is paucity of research studies evaluating the cost-effectiveness of HCBIs of MI and HC. Likewise, very few available studies have clearly stipulated how HCBIs can reduce expenditure. Future research needs to examine the cost-effective aspect when involving MI/HC strategies in behaviour change interventions to promote oral health.

### 4.6. Research Gaps

This scoping review identified research gaps in the literature on health coaching-based interventions for oral health improvement.

I.No paper or article included in this review addressed the role of organisational, regulatory, fiscal, or legislation as factors directly influencing interventions and oral health delivery and outcomes.II.There is a limited number of studies using the design of randomised controlled trials [30,31,32,34,40,42,43,44,45,48] showing varied success of the interventions in improving oral health. However, more studies with methodological rigor targeting at various age groups and behaviours are required to better understand their roles in oral disease management.III.Only two studies [35,47] explored the role of the interventions in changing dietary habits, which is one of the crucial issues for the oral health behaviour change. However, further studies are required in all other age groups.IV.Only two studies addressed the role of plaque control measures, such as toothbrushing in improving oral health outcomes and oral health behaviour change [32,47]. López-Jornet et al. [32] showed how the frequency and use of interproximal brushing affect plaque and bleeding indices, while Cinar et al. [47] explored how toothbrushing efficacy and frequency affect oral health and quality of life among patients with type 2 diabetes.

### 4.7. Limitations

Some of the articles [30,34,43,44,46] included in this scoping review were also reviewed in Werner et al., [2] systematic review. This maybe one of the limitations of this study, however this issue was addressed during theme extraction phase where we critically analysed each of those included studies to extract the final recommendations and conclusions. A second limitation in this scoping review report is the inclusion criteria, as the included articles were written in English only. Another limitation was the scoping review design. Compared to systematic reviews, scoping reviews are considered less rigorous; they provide a descriptive account of available information and do not formally evaluate the quality of evidence. The search strategy was another limitation, as article selection was limited to online sources.

## 5. Conclusions

This review has mapped out how HCBIs research is conducted and has highlighted the research gaps, suggesting a need for further studies linking interventions to legislation, costs and planning. Results from this study should be taken with caution as more studies covering wider populations with varying important dental age groups are required to better understand the role of these interventions in oral disease management and oral health promotion. Nonetheless, the extracted characteristics suggest that health coaching-based interventions can be effective if (i) oral health professionals receive appropriate training in the use of MI/HC strategies for patient behaviour change and can engage patients by avoiding criticism during the behaviour change process and (ii) provide routine brief HC/MI sessions in clinical settings, (iii) oral health professionals can supplement traditional health education methods with individually tailored communication when delivering behaviour change sessions and (iv) use MI/HC strategies for cost-effectiveness purposes.

## Figures and Tables

**Figure 1 dentistry-11-00073-f001:**
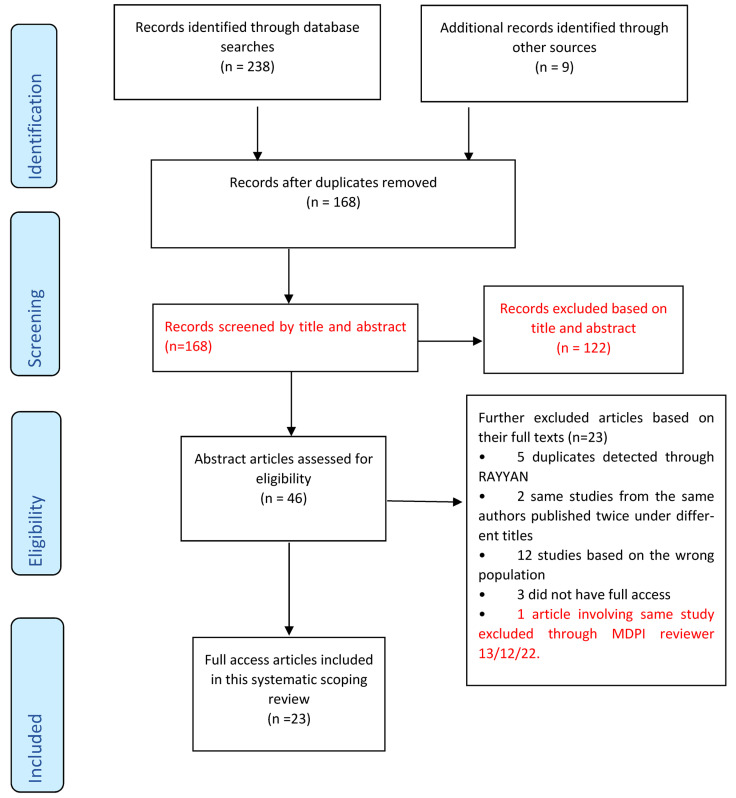
PRISMA-ScR flow chart adapted from Tricco et al. [22].

**Table 1 dentistry-11-00073-t001:** A summary of eligibility criteria.

PCC Framework	Inclusion Criteria	Exclusion Criteria
Participants	Adult patients aged 18 years or over	Studies including children, or those with cognitive impairments.
Concept	Interventions that have a component of oral health improvement using health coaching (HC) or motivational interviewing (MI) techniques to encourage participants’ oral health behaviour or lifestyle change.	Interventions with no oral health promotion elements or no communication techniques to support participants in oral health-related behaviour change.
Context	Community and clinical settings	Studies not based on community/clinical settings
Language	English	Non-English

## Data Availability

The protocol for this review was registered with the Open Science Framework and can be accessed from the following web page: https://osf.io/kbcjt.

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
