# Peer review of "Health Coaching-Based Interventions for Oral Health Promotion: A Scoping Review"

_dentistry, 2023, doi:10.3390/dj11030073_

Round 1

Reviewer 1 Report

First, I would like to congratulate the authors. The review highlights a significant and current issue, quite relevant in modern dentistry, which makes the article valuable.

The manuscript is clear and valid, the background is short but introduces the subject in a clear and objective way.

 Materials and Methods are described in detail without errors and with logic. The methodology respects the steps for conducting a Scoping Review

Additionally results and discussion are well written and made in an objective way. The reading of the tables is easy, useful and understandable.

The following comment is addressed to enhance the quality of the manuscript, and could be added in introduction:

Scoping reviews are typically less exhaustive than systematic reviews, what was the authors purpose of conducting a scoping review instead of a systematic review?

Author Response

Many thanks, please see the attachment

Reviewer 2 Report

Comments to the Author

Thank you very much for the opportunity to review the article.

This scoping review aimed to synthesize evidences by reviewing Health Coaching-Based Intervention Strategies (HCBIS) comprehensively for oral health promotion in the community and clinical settings. The authors extracted data from 24 included articles by literature search. Study designs of included articles are RCT, quasi-experimental/interventional study, and systematic review. The authors concluded that the effect of the motivational interviewing (MI) was overall significant to improve behavioral changes and oral health, yet magnitude of effect varied by studies.

<Methods>

ž   Please check the spelling of RYYAN (line 91).

ž   Please add manufacturer’s information for EndNote and Rayyan

ž   The reviewer cannot understand the reason of including systematic reviews and RCTs. Evidences from these studies are difficult to synthesize. Please describe reasons of including these completely different study designs in this review.

ž   Please explain the reason choosing scoping review, not systematic review.

<Results>

ž   Tables can be improved as they (especially table 1 and 4) are too big to catch the meaning.

ž   67 articles were excluded by 2nd reviewer. Please provide detailed reasons in figure 1.

ž   In table 2, study population of some studies are adult patients. Please clarify this point as one of the exclusion criteria is chronically ill patients.

ž   In table 2, some information is redundant. If articles use t-test for statistical analysis, it is apparent that they are quantitative studies.

ž   The authors mentioned that quality of included studies were evaluated by using JBI critical appraisal tools. However, there are no detailed information provided regarding low/medium/high evaluation. It is helpful if authors provide this information as supplemental files.

ž   Reference 14, 21 were included for the scoping review. These seem to be based on an identical study, and the study population were patients of diabetes type 2.

<Discussion>

Please add reference numbers when discussing the key findings in 4.1, 4.2, 4.3, 4.4, and 4.5. For example in 4.1, “The finding on engaging clients or patients and avoiding negative feedback and criticism in this review mirrors the work of Wong-Rieger and Rieger [34]”. The reviewer cannot follow the discussion, as the authors does not provide any reference numbers of included articles.

Author Response

Many thanks, please see the attachment

Reviewer 3 Report

The article looks well organized. Article can be published under the editor decision.

Author Response

Many thanks, please see the attachment.

Round 2

Reviewer 2 Report

Thank you very much for giving me a chance to review an article. The manuscript has substantially improved. 

Minor comment: Please add reference numbers to included articles in the table 4.

Major comment: I have one concern about overestimation of the result.  In this study, included studies are systematic reviews and original researches.  Some of original studied reviewed by this scoping review were also reviewed by included systematic reviews. For example, the authors included Werner Hakeberg et al.'s article (reference no 2). Some original studied reviewed by Werner et al. (Godard et al., Brand et al., Stenman et al., Jonsson et al. ) were also included in this scoping review.  Did authors consider this redundancy in the analysis? If so, please mention this point in the material and methods section. Otherwise, please explain this in the limitation. 

Author Response

Many thanks, please see attachment
